# Follicular Thyroid Adenoma and Follicular Thyroid Carcinoma—A Common or Distinct Background? Loss of Heterozygosity in Comprehensive Microarray Study

**DOI:** 10.3390/cancers15030638

**Published:** 2023-01-19

**Authors:** Martyna Borowczyk, Paula Dobosz, Ewelina Szczepanek-Parulska, Bartłomiej Budny, Szymon Dębicki, Dorota Filipowicz, Elżbieta Wrotkowska, Michalina Oszywa, Frederik A. Verburg, Małgorzata Janicka-Jedyńska, Katarzyna Ziemnicka, Marek Ruchała

**Affiliations:** 1Department of Endocrinology, Metabolism and Internal Medicine, Poznan University of Medical Sciences, 60-355 Poznan, Poland; 2Department of Medical Simulation, Poznan University of Medical Sciences, 60-806 Poznan, Poland; 3Department of Genetics and Genomics, Central Clinical Hospital of the Ministry of Interior Affairs and Administration, 02-507 Warsaw, Poland; 4Department of Radiology and Nuclear Medicine, Erasmus Medical Center, 3015 GD Rotterdam, The Netherlands; 5Department of Clinical Pathology, Poznan University of Medical Sciences, 60-355 Poznan, Poland

**Keywords:** follicular thyroid adenoma, follicular thyroid cancer, genetics, loss of heterozygosity, microarray, molecular inversion probe

## Abstract

**Simple Summary:**

Approximately 50% of 60-year-old persons have thyroid nodules that in 7–15% may be thyroid cancer. Diagnosis of follicular thyroid adenoma (FTA) and follicular thyroid cancer (FTC) is particularly challenging. Furthermore, it is not clear whether they share a common or distinct background. The study aimed to compare FTA and FTC using the comprehensive microarray for the first time and to identify recurrent regions of loss of heterozygosity. We found that FTA and FTC may share a common genetic background—including the same LOH in 16p12.1, which encompasses many cancer-related genes. However, differentiating rearrangements may also be detected, such as LOH in 11p11.2-p11.12 only in FTA patients (56% vs. 0%) and LOH in 12q24.11-q24.13 detected more often in FTC (37.5% vs. 6.3% in FTA). Genomic screening may show the complexity of genetic background in follicular thyroid lesions and enable the identification of new genetic rearrangements participating in FTC pathogenesis.

**Abstract:**

Pre- and postsurgical differentiation between follicular thyroid adenoma (FTA) and follicular thyroid cancer (FTC) represents a significant diagnostic challenge. Furthermore, it remains unclear whether they share a common or distinct background and what the mechanisms underlying follicular thyroid lesions malignancy are. The study aimed to compare FTA and FTC by the comprehensive microarray and to identify recurrent regions of loss of heterozygosity (LOH). We analyzed formalin-fixed paraffin-embedded (FFPE) samples acquired from 32 Caucasian patients diagnosed with FTA (16) and FTC (16). We used the OncoScan™ microarray assay (Affymetrix, USA), using highly multiplexed molecular inversion probes for single nucleotide polymorphism (SNP). The total number of LOH was higher in FTC compared with FTA (18 vs. 15). The most common LOH present in 21 cases, in both FTA (10 cases) and FTC (11 cases), was 16p12.1, which encompasses many cancer-related genes, such as *TP53*, and was followed by 3p21.31. The only LOH present exclusively in FTA patients (56% vs. 0%) was 11p11.2-p11.12. The alteration which tended to be detected more often in FTC (6 vs. 1 in FTA) was 12q24.11-q24.13 overlapping *FOXN4*, MYL2, *PTPN11* genes. FTA and FTC may share a common genetic background, even though differentiating rearrangements may also be detected.

## 1. Introduction

Approximately 50 percent of 60-year-old persons have thyroid nodules [1]. Their clinical importance relates to the need to exclude thyroid cancer [2], which occurs in 7%–15% of cases [3]. The differentiation between benign and malignant thyroid lesions is usually based on ultrasound and thyroid fine-needle aspiration biopsy [4]. One particular type of thyroid nodule—follicular thyroid lesion including follicular thyroid adenoma (FTA) and follicular thyroid cancer (FTC)—remains a diagnostic challenge [5,6]. Their presurgical differentiation is not reliable when based on imaging or cytology [7]. Furthermore, their natural history is not clear—do they share a common or distinct background? Is FTC an extension of FTA? What is the phenomenon underlying malignancy and invasion of FTC?

In recent years, many studies have endeavored to identify additional factors to discriminate between those two related pathological entities [8] and to further clarify the relationship’s extent in their pathogenesis [9].

Typical markers of malignancy may appear as specific genetic alterations that may be present in FTC but not in FTA, especially since molecular markers are increasingly used as presurgical diagnostic tools in the management of indeterminate thyroid nodules [10]. However, the results of molecular testing performance of cytology specimens from thyroid nodules are still not fully satisfactory [11].

Although next-generation sequencing (NGS) has generated promising results in papillary thyroid cancer (PTC) [12], this technique has rarely been used in FTC [13] thus far. Moreover, although the genomic landscape of PTC is nearly complete, the molecular characterization of FTC and its progression from minimally invasive FTC to widely invasive FTC are still not totally clear [14]. Recently, our study group has proposed a few genetic markers of thyroid follicular lesions’ malignancy based on our NGS studies [15,16,17]. However, NGS is not exhaustive, as it only yields information about single point mutations. 

To have a comprehensive insight into follicular thyroid lesions’ genetic landscape, analysis of larger rearrangements is needed. Recent array technology allows whole-genome screening, including copy number variants (CNVs) analysis at a high resolution. The microarray can determine chromosomal abnormalities and genomic instability; i.e., highly accurate pathogenic copy number variants and allelic imbalances in solid tumors from limited amounts of DNA [18]. Thus far, many studies have included only a targeted analysis of CNVs and loss of heterozygosity (LOH) of FTA or FTC [19,20,21]. The latter has been regarded as a characteristic feature of the follicular phenotype [19].

The OncoScan™ microarray assay (Affymetrix/Thermo Fisher Scientific, Waltham, MA, USA), which uses highly multiplexed molecular inversion probes for single nucleotide polymorphism (SNP) loci, has been developed and validated for genomic profiling of somatic copy number aberrations of various tumors [22,23]. To the best of our knowledge, researchers have not yet used this technique to simultaneously detect many genetic changes and analyze the complex interplay of distinct genetic alterations. 

The aim of the study was to compare FTA and FTC using the high-resolution SNP array for the first time and to identify recurrent regions of LOH, which may support preoperative differentiation and a better understanding of those entities.

## 2. Materials and Methods

We analyzed 32 randomly selected patients at the Department of Endocrinology of our university hospital, who were diagnosed with follicular lesions (16 with FTA and 16 with FTC). The diagnosis was made according to the World Health Organization criteria [24]. We adjusted both groups for age and gender. The Bioethical Committee of Poznan University of Medical Sciences approved the study (approval no. 1061/15, January 2015), and it was conducted in accordance with the Declaration of Helsinki [25]. Due to the retrospective nature of the analysis and the use of stored materials, additional informed consent was not required. 

We subjected formalin-fixed paraffin-embedded (FFPE) samples obtained from total or subtotal thyroidectomy, and we gained the detailed clinical annotates. The study group included 28 women and 4 men with a median age of diagnosis of 55 years (range: 29 to 82). All patients were Caucasian and did not suffer from any other endocrine disorders or cancers. They were not receiving any treatment at the time of diagnosis. The analysis covered data collected between 2008 and 2020. Patient characteristics are presented in Table 1.

The specimens were acquired by thyroidectomy and re-examined by a qualified pathologist, who confirmed the diagnosis of FTA and FTC. Patient and sample data included: the age at diagnosis, gender, tumor size, multifocality (when two or more foci were found), extra-thyroidal extension, the presence of histopathological signs of chronic lymphocytic thyroiditis, histopathological staging (pTNM) according to the 8th tumor-node-metastasis (TNM) classification [24], and radioiodine refractoriness (if a cumulative activity of 22.2 GBq/600 mCi of RAI did not result in achieving complete remission). All samples were anonymized.

A qualified pathologist indicated areas of interest from FFPE specimens, and slides were manually microdissected. Genomic DNA was extracted using a QIAamp DNA FFPE Tissue Kit (Qiagen, Valencia, CA, USA), following the manufacturer’s instructions. Genomic DNA was quantified using a fluorometer Qubit platform (Invitrogen, Carlsbad, CA, USA), and the DNA quality and integrity were tested.

Further studies included whole-genome microarrays based on molecular inversion probes performed with the use of OncoScan™ arrays (Affymetrix, Thermo Fisher Scientific, Waltham, MA, USA). All procedures related to genomic screening were conducted according to the array provider’s instructions to ensure proper efficiency and reliability of results. This method enabled analysis of copy number changes and allelic imbalances across the entire genome.

To minimize the number of false-positive findings, the following criteria for significant results were followed: genomic deletions of a minimum region of 150 kb and genomic amplifications of a minimum region of 200 kb.

To calculate the copy number of altered regions, the data were normalized to baseline reference intensities using NA 32.3v.2 reference model (provided by Affymetrix/Thermo Fisher Scientific, Waltham, MA, USA). The Hidden Markov Model (HMM) available within the software package was used to determine the copy number states and their breakpoints. Thresholds of log2ratio ≥ 0.58 and ≤1 allowed to categorize altered regions as CNV gains (amplifications) and copy number losses (deletions), respectively. To prevent the detection of false positive CNVs, at least 50 consecutive, adjacent probes were considered in the recognition of gains or losses in our study. Gains or losses were analyzed separately. To exclude aberrations representing common normal CNVs, all the identified CNVs were compared with those reported in the Database of Genomic Variants (DGV, http://projects.tcag.ca/variation/, accessed on 30 April 2022). To identify the genes within the CNVs, we used the UCSC database (http://genome.ucsc.edu, accessed on 30 April 2022) and Ensemble (http://www.ensembl.org, accessed on 30 April 2022). Gene annotation and gene overlap were assigned using the human genome build 19 (hg19) and NetAffx (http://www.affymetrix.com, accessed on 30 April 2022). In addition, the identified alterations were confronted with data deposited in COSMIC database (www.http://cancer.sanger.ac.uk, accessed on 30 April 2022) to look for overlap with up-to-date known genomic cancer regions and genes.

The algorithm for detection of copy number aberrations in tumor cell mixture (mosaicism and clonality) considered comprehensive analysis of adjacent single copy deletions and gains segments. The algorithm was designed to be most accurate when the normal/expected copy number state (CN) is diploid and targets detection of changes in regions of approximately 5Mb or more in size and variation within min. 500 markers (being typical for segments of 5000 markers or larger in size). This approach considers only a discrete number of mosaicism levels, which are set at 30%, 50%, and 70%. The range of log ratios is broken into a series of bands according to the detection level (30% or greater, 50% or greater, and 70–100% bands). This tool was most efficient in detecting mosaicism between approximately 30–70% of cells and for copy numbers between 1 and 3. The LOH were used for the final analysis. 

The data obtained from genomic experiments were analyzed using a dedicated OncoScan™ Console 1.3 and ChAS v4 software and were compared with clinical data.

In order to fully explore identified alterations and contributing genes, we used a bioinformatics approach and tools such as DVID (The Database for Annotation, Visualization and Integrated Discovery) and Ingenuity^®^ Pathway Analysis (IPA^®^, Qiagen Sciences, Germantown, MD, USA). 

Parameters were recorded and entered into a dedicated database. Descriptive analysis was used to summarize the collected data. To determine the normality of continuous variables, data were tested by the D’Agostino and Pearson omnibus normality test. Variables that were found to be normally distributed were expressed as mean values. Data that were found to be distributed differently were expressed as median and minimum–maximum values.

To compare differences between the groups for categorical variables, the chi-square test was used if the Cochrane assumptions were met; otherwise, Fisher’s exact test was used. Interval data were compared using the Mann–Whitney U test as the data did not follow a normal distribution.

A *p*-value of less than 0.05 was regarded as significant. Statistical analyses were performed with StatSoft Statistica v13.0 and PQStat v1.6.8 software.

## 3. Results

The total number of LOH was higher in FTC compared with FTA (18 vs. 15). The most common LOH present in 21 cases, including both follicular thyroid adenoma (10 cases) and follicular thyroid carcinoma (11 cases), was 16p12.1 comprising over 7.5 Mbp in size, encompassing 149 known genes, including many important cancer-related genes such as TP53 and its variants, *UBE2MP1*, *ATP2A1*, *IL27*, *TGFB*, *MAPK3*, *BCL7C*, and many transcription factor subunit genes. Another LOH present in both types of thyroid cancer was 3p21.31, about 6.4 Mbp in size, comprising 172 genes, including *KIF9, SLC26A6*, *UBA7*, *CACNA2D2*, *TLR9*, and *BAP1*. This region was affected in nine cases of FTC and seven cases of FTA. Moreover, LOH 15q15.1 was frequently found in both follicular lesions; it comprises over 3.6 Mbp and includes genes such as *TYRO3*, *CAPN3*, *TP53BP1*, and *EIF3J*.

The only LOH present exclusively in FTA patients (56% vs. 0%) was 11p11.2-p11.12 (5.4 Mb in size), including *KAI1*, a metastasis suppressor gene, *OR4* subfamily genes and *CREB* family transcription factor members. Another LOH on chromosome 20 (q11.21-q11.23, 6.89 Mb) also predominated in FTA and consisted of genes such as *SRC*, *BCL2L*, *DNMT3B*, and *MMP24*, among others. Only one patient with thyroid cancer different to FTA presented this LOH region. 

The alteration which tended to be detected more often in FTC was 12q24.11-q24.13 (region size: 3.99 Mb) overlapping *FOXN4*, *MYL2*, *PTPN11*, *UBE3B*, *RAD9B*, and *RASAL1* genes, as well as *OAS* gene family. This region was affected in six cases of FTC, but only once among the FTA cohort. 

Table 2 presents all LOHs present in both types of lesions, and predominantly in one type of thyroid cancer, including genes detected in the LOH region, together with non-coding RNA genes.

Other significant LOHs detected only in FTC patients, albeit less frequently, included the following regions: 1q21.1 (about 3.03 Mbp, most important genes: *PDE4DIP*, *BCL9*), 2q11.2 (over 4.2 Mbp, including *ARID5A* and *COX5B* genes), 3p12.2 (about 3.7 Mbp, including *GBE1* gene), 8q11.1 (over 2.8 Mbp in size, including *MCM4* and *UBE2* genes), 14q23.3 (3.5 Mbp, including *MAX*, *ATP6*, *EIF2*, *ARG2*, *RAD51B*, and *GPHN* genes), 14q32.31 (over 2.8 Mbp, including *HSP90*, *TRAF3*, *TNFA*, *APOPT1*, and *KIF26A*), and 22q11.23 (0.82 Mbp, including *GSTT1*, *GSTT2,* and *CABIN1* genes). Even though most of the changes mentioned above have been of “loss” type, several “gain” changes have been also discovered, especially in the following regions: 19q13.41 (nearly 2 Mbp in size, including *ZNF331*, *PPP2R1A*, *HAS1*, and *BIRC8*), 1p13.3 (44,148 in size, including *GSTM* gene family members), 3p14.1 (0.43 Mbp, including *SUCLG2* and *FAM19A1* genes), and 13q12.3 (about 2.7 Mbp in size, including *BRCA2* gene). 

Regarding the differences in a group of FTC patients, we found an increased frequency of LOH in the 3p21.31 for the T2–T3 group (*p* = 0.001). Nodal involvement has been accompanied by 14q32.31 LOH. In a case of capsular invasion, the most common were LOHs in the 12q24.11 and 16p12.1. Multifocality was linked to 2q11.2 and 8q11.1.

Among FTA patients, other detected LOHs included: 1q21.1 (over 2.5 Mbp, including *PDE4DIP* and *CD160* genes), 3p12.2 (over 3.7 Mbp, including *GBE1* gene), 3q28 (nearly 6.7 Mbp, including *TFRC*, *IL1RAP*, and *FGF12* genes), 7q11.21 (nearly 3.8 Mbp, including *SBDS* and *VKORC1* genes), 8q11.1 (2.8 Mbp, including *PRKDC* gene), 9q34.3 (1.8 Mbp, including *NOTCH1* gene), 10q11.21 (3.1 Mbp, including *NPY4R* and *AGAP9*), 17q21.31 (nearly 3kbp, including *MAP3K*, *MAPT*, *NSF*, and *WNT3*), 20q13.33 (0.15 Mbp, including *HRH3* and *SS18L1*), and 14q32.31 (4.2 Mbp in size, including *AKT1* and *JAG2*). Even though most of the changes mentioned above have been of “loss” type, several “gain” changes have been also discovered, especially in the regions 1p13.3 and 3p14.4, although no significant genes related to the cancer process have been reported in those areas. Figure 1 shows the results of the comparison analysis. We identified which markers are significantly different between FTA and FTC and which are similar, and we compared them with known ones from the previous research.

## 4. Discussion

The results indicate that FTA and FTC may share a common genetic background, even though differentiating rearrangements may also be detected. The most common LOH region, 16p12.1, is present in both FTA and FTC, with similar numbers of cases. Also, 3p21.31, as well as 15q15.1 deletion. These big regions contain several pseudogenes and genes with non-coding RNA products, but also many important genes, including those with known involvement in cancer pathogenesis, as indicated in Appendix A. The same LOH present in both FTA and FTC localized on 3p has been described already in 2008. In the study of Hu et al., LOHs on chromosome 3p, including 3p21, were detected in 71% of follicular thyroid carcinoma (17/24), 30% of papillary thyroid carcinoma (9/30), and 10% of follicular adenoma (2/20) cases [32]. The concept of FTA and FTC similarities is not new as RAS somatic mutations and PAX8/PPARγ rearrangements, the key FTC alterations, were detected in both FTCs and FTAs [33].

LOH 12q24.11, present primarily in follicular thyroid carcinoma, may constitute a possible marker of follicular thyroid lesions’ malignancy as it seems to include genes strictly associated with thyroid cancer pathogenesis, as shown in Appendix A. LOH in 12q24 has been regarded as potentially pathogenic for gastric cancer [34] and for neuroblastoma progression [35]. Locus 12q24.11 encompasses the human glycolipid transfer protein (GLTP) gene responsible for glycosphingolipid metabolism, including lipid binding and glycolipid transfer activity [36]. Another LOH present only in FTC was allelic deletions in 22q, previously described in highly invasive FTCs with poor prognosis [37]. 

On the other hand, follicular thyroid adenoma’s most specific regions, 11p11.2 and 20q11.21 LOH, comprises several gene encoding transcription factors and many miRNA genes, which are also known for their impact on carcinogenesis, as well as direct and indirect regulation of the transcription factor function. Further, LOH within the short arm of chromosome 11 (11 p) in the development of FTA has been previously described [38]. Selected genes have been depicted in Appendix A.

Overall frequency of allelic loss (OFAL), including LOH, has been regarded as a characteristic feature of the follicular phenotype [19,39]. It is why we decided to precisely analyze LOH in FTA and FTC in our study. They represent molecular disorders acquired by the cell during neoplasm transformation [20]. In this disorder, one of the gene alleles is lost in a neoplastic cell [40]. The mechanism of LOH is chromosome-specific. Knudson et al. [41] postulated that loss of function of both tumor suppressor gene (TSG) copies is required for an uncontrolled proliferation of modified cells and, eventually, for neoplastic transformation [20]. One allele may be inactivated by promoter hypermethylation or point mutation (e.g., substitution) or intragenic microdeletion, while the second allele may be lost via LOH [41]. It was confirmed by a finding of LOH high percentage in anaplastic thyroid carcinoma, suggesting that LOH may be regarded as a late event in thyroid tumorigenesis associated with the loss of tumor differentiation and increased degree of aggressiveness [20].

The similarities between the FTA and FTC genetic backgrounds found in our study supports the hypothesis that they may constitute a partial continuum in their natural history. This was also previously suggested by Nikitski et al. [42] and many others, who analyzed TP53-mutant FTA as a precursor not only to FTC, but also to anaplastic thyroid carcinoma. In our study, deletion of a region including *TP53BP1* gene was present in both FTA and FTC.

The new LOHs may occur not only as a sign of transformation from FTA and FTC, but also of its growth. In their study, Migdalska-Sęk et al. [19] found regions with significantly increased frequency of LOH/MSI for specific histotypes: the 3p24.2 region for FA and 1p31.2 for FTC. LOH/MSI in 3p21.3 was significantly elevated in PTC and FTC. LOH/MSI in 3p21.3 was increased for small-size tumors (T1a + T1b), tumors with no regional lymph node involvement (N0 + Nx), American Joint Committee on Cancer (AJCC) stage I tumors, and tumor diameter (Td) < 10 mm. We confirmed the occurrence of LOH3p21.31 in both FTA and FTC, indicating early-stage tumorigenesis.

Similarly to the results of our study, the number of allelic losses (LOH) calculated in different studies increases from the lowest in FTA to higher numbers in FTC, with the highest number for anaplastic thyroid cancer [27,38,43]. It was also higher for atypical FTA than for typical FTA [44]. This increased number of LOH events may contribute to the clinical aggressiveness of cancer [45]. The follicular adenoma–carcinoma sequence in thyroid carcinogenesis may include atypical follicular thyroid adenoma as an important intermediate in this pathway [46]. The results of our study also prove that the incidence of LOH may overtly increase with tumor progression. The incidence of LOH and the numbers of loci, in which the loss of heterozygosity increases with the degree of neoplastic progression, indicates a successive accumulation of molecular disorders in cells and a coincidence of LOH and mutations in thyroid tumorigenesis.

Our study demonstrates that the role of LOH in the process of thyroid neoplastic transformation lays in an inactivation of various genes by deletions in their loci, which is already present in preneoplastic lesions. It may prove the role of LOH in the process of carcinogenesis initiation and neoplastic transformation of the thyroid gland. It is compatible with the classical multistep carcinogenesis model applied to FTC, which is based on a theory of cancer clonal origin. Genome instability within somatic cells is the first step. Then, more aggressive clones appear, and they survive the selection pressure of the microenvironment. The appearance of a functionally significant mutation leads to divergence of a new subclone, which can dominate and outcompete other cells to result in a homogenous tumor until a new, significant, and more versatile mutation appears [47]. According to this theory, FTC is considered to be derived from FTA [48].

One of the hot LOH sites included in the first step may be 15q region, while in our study its LOH were present both in FTA and FTC. It may lead to a neoplastic transformation of normal thyroid cells towards adenoma. In turn, the imprinted genes take part in differentiation of typical adenomas into an atypical form, thus confirming their role in an early stage of neoplastic transformation towards malignant lesion (FTC) [49,50].

Loss of heterozygosity in the 3p21.31, 15q15.1, or 16p12.1, present in both FTA and FTC, may be regarded as an early and probably initiating event in the development of follicular adenoma. The progressive character of the process is particularly evident for LOH in the 12q24.11 locus, which may have a role in FTA transformation towards FTC. The high incidence of the loss of heterozygosity in FTC (37.5%) and lower incidence in FTA (6.25%) suggest that the 12q24.11 locus could be the minimally deleted regions (MDR), specifically for FTC.

Moreover, the higher percentage of malignant tumors with LOH observed in those loci vs. benign lesions may confirm the hypothesis that the development of FTA and FTC is associated with a clonal event, occurring in a single precursor cell.

We hypothesize that the 16p12.1, 3p21.31, or 15q15.1 deletion sensitizes the genome for disease, while “second-hits” in the genetic background, such as 12q24.11 deletion, modulate the phenotypic trajectory and cause tumorigenesis. The same role of 16p12.1 deletion as “the first hit” was suggested for maldevelopment of nervous system. 

The results of our study are passed on molecular inversion probe (MIP) technology. The analysis of LOH is comprehensive as MIP is a proven technology for identifying rearrangements and simultaneous detection of selected somatic mutations (so-called “driving mutations”). This assay has been shown to perform well with low inputs of DNA starting material, making the assay a natural choice in cancer clinical research. It enabled having FFPE as a source of DNA, despite a DNA degradation and getting rid of method bias, as the OncoScan^TM^ algorithms have been especially developed to address two major challenges associated with solid tumor copy number analysis: first, establishing the expected normal copy number state for a given locus, and second, accounting for “normal cell contamination” present in most samples, which affects copy number estimates.

The study has two main limitations. The first limitation is the small number of participants. However, to the best of our knowledge, the method used in our study (highly multiplexed molecular inversion probes for SNP loci) has been used for the first time in FTA and FTC analysis. Contrary to previous research (also including a small number of patients), we were able to simultaneously detect many genetic changes. The second one is a lack of any experiments about the expression of mapping genes to understand if the allelic copy still present in FTC samples is inactivated or functional. It requires a further analysis by in situ hybridization or immunohistochemistry on the same FFPE specimens to understand if the LOH is simply an FTC marker or if they represent potential tumor suppressors in FTC. We consider our study as a starting point for future research increasing the number of FFPE specimens from additional patients and including functional analysis.

Genomic screening may show the complexity of follicular thyroid lesions’ genetic background and enable the identification of new genetic rearrangements participating in FTC pathogenesis. Given the general similarities of FTA and FTC and the same tissue origin, some LOH differences may reflect malignant progression potential, including useful candidate biomarkers for FTC and identifying factors important for FTC pathogenesis.

## 5. Conclusions

The results indicate that FTA and FTC may share a common genetic background, even though differentiating rearrangements might also be detected. 12q24.11 LOH may constitute a possible marker of malignancy as it includes genes strictly associated with thyroid cancer pathogenesis. Genomic screening may show the complexity of follicular thyroid lesions’ genetic background and enable the identification of new genetic rearrangements participating in FTC pathogenesis.

## Figures and Tables

**Figure 1 cancers-15-00638-f001:**
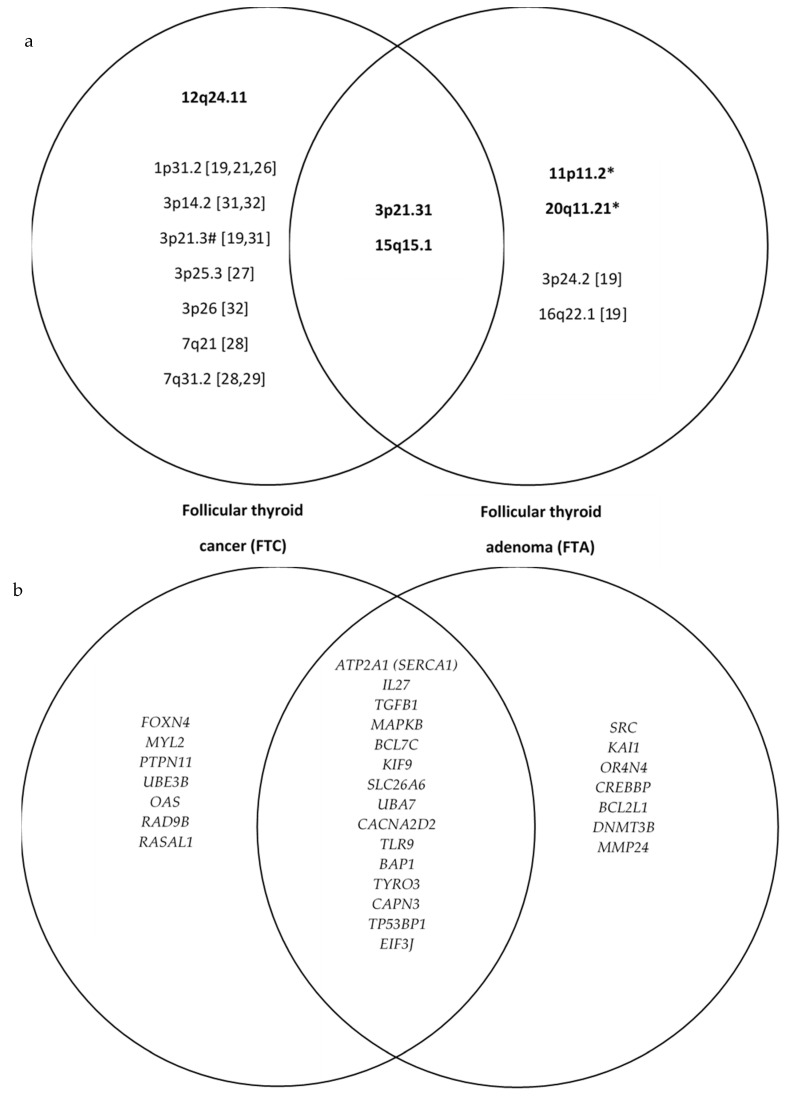
(**a**) The overlapping chromosomal regions with loss of heterozygosity in follicular thyroid cancer (FTC) and follicular thyroid adenoma (FTA). The first set shows LOH present only in follicular thyroid cancer, the second both in FTC and FTA, and the third shows LOH present only in FTA. Findings from the present study are in bold. Other regions (not in bold) were found in the literature [19,21,26,27,28,29,30,31,32]. (**b**) The most important genes included in overlapping chromosomal regions with loss of heterozygosity in follicular thyroid cancer (FTC) and follicular thyroid adenoma (FTA). * represents statistical significance at *p* < 0.05; # represents regions present only in FTC or in both FTC and FTA (the latter in our study).

**Table 1 cancers-15-00638-t001:** Patient characteristics according to histopathological diagnosis.

Characteristics	Follicular Thyroid Adenomas *n* = 16	Follicular Thyroid Carcinomas *n* = 16	*p*-Value ^1^
Male/female, *n* (%)	2/14 (12.5%/87.5%)	2/14 (12.5%/87.5%)	1.000
Median age at diagnosis, years (range)	53 (29–81)	56 (31-82)	0.649
Age group ≤60 years/>60 years), *n* (%)	11/5 (68.8%/31.2%)	10/6 (62.5%/37.5%)	1.000
Median length of follow-up, months (range)	119 (58–162)	152 (47–174)	0.587
Multifocality, *n* (%)	0	2 (12.5%)	0.4839
Capsule invasion, *n* (%)	NA	7 (43.8%)	NA
Extracapsular extension, *n* (%)	NA	10 (62.5%)	NA
Nodal (N) involvement, *n* (%)	NA	1 (6.3%)	NA
Mean tumor size, mm (range)	23 (6–50)	26 (8-50)	0.112
Tumor diameter ≤10 mm, *n* (%)	3 (18.8%)	1 (6.3%)	0.5996
Localization in the right/left/both lobes, *n* (%)	9/7/0 (56.3%/43.7%)	8/7/1 (50%/43.8%/6.2%)	0.7222
Chronic lymphocytic thyroiditis, *n* (%)	2 (12.5%)	3 (18.8%)	1.000
Radioactive iodine refractoriness *n* (%)	NA	1 (6.3%)	NA

^1^ The *p*-values were based on a chi-square test (or Fisher’s exact test where appropriate) for categorical variables and a Mann–Whitney U test for quantitative variables. NA—not applicable.

**Table 2 cancers-15-00638-t002:** LOHs present in both types of lesions or predominantly in FTA or FTC. Census genes detected in the LOH region are listed.

Chrom.	Cytoband Start	Size (kbp)	Gene Count	Census Genes	Microarray Nomenclature	FTC	FTA	Sum	*p*-Value	OR (95% CI)
LOHs present in both follicular thyroid carcinoma and follicular thyroid adenoma										
16	p12.1	7500.913	149	*FUS*	arr[hg19] 16p12.1-p11.1(27,770,812–35,271,725) hmz	11	10	21	0.710	1.32 (0.31–5.70)
3	p21.31	6391.659	172	*SETD2*, *NCKIPSD*, *RHOA*, *BAP1*, *PBRM1*	arr[hg19] 3p21.31-p21.1(46,778,841–53,170,500) hmz	9	7	16	0.481	1.65 (0.41–6.68)
15	q15.1	3616.641	70	*B2M*	arr[hg19] 15q15.1-q21.1(41,796,900–45,413,541) hmz	9	5	14	0.159	2.82 (0.67–12.02)
LOHs present predominantly in follicular thyroid carcinoma:										
12	q24.11	3990.65	59	*SH2B3*, *ALDH2*, *PTPN11*	arr[hg19] 12q24.11-q24.13(109,669,669–113,660,319) hmz	6	1	7	0.057	9.00 (0.94–86.53)
LOHs present predominantly in follicular thyroid adenoma:										
11	p11.2	5404.548	58	*CREB3L1*, *DDB2*	arr[hg19] 11p11.2-p11.12(46,171,403–51,575,951) hmz	0	9	9	0.014	41.80 (2.14–816.37)
20	q11.21	6885.552	125	*ASXL1*, *SRC*	arr[hg19] 20q11.21-q11.23(29,519,155–36,404,707) hmz	1	5	6	0.099	6.82 (0.69–66.91)

## Data Availability

The data presented in this study are available on request from the corresponding author. The data are not publicly available due to privacy and ethical restrictions.

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
