# Peer review of "Follicular Thyroid Adenoma and Follicular Thyroid Carcinoma—A Common or Distinct Background? Loss of Heterozygosity in Comprehensive Microarray Study"

_cancers, 2023, doi:10.3390/cancers15030638_

Round 1

Reviewer 1 Report

This paper describes the results of loss of heterozygosity in formalin-fixed paraffin embedded materials of 16 follicular thyroid adenoma (FTA) and 16 follicular thyroid carcinoma (FTC) by OncoScan microarray assay system. The paper is very interesting because it shows the LOH common to follicular adenoma and follicular carcinoma, the LOH specific to each, and the candidate genes in those regions. The results of this paper are useful in understanding the genetic background of follicular tumors, and the previous literature is well cited and comprehensive.

Only the following point need to be added to Results and Discussion. Regarding LOH in follicular carcinoma, differences by tumor size, presence of capsular invasion, multifocality, and nodal involvement should be shown.

Author Response

Reviewer #1:

Comment: This paper describes the results of loss of heterozygosity in formalin-fixed paraffin embedded materials of 16 follicular thyroid adenoma (FTA) and 16 follicular thyroid carcinoma (FTC) by OncoScan microarray assay system. The paper is very interesting because it shows the LOH common to follicular adenoma and follicular carcinoma, the LOH specific to each, and the candidate genes in those regions. The results of this paper are useful in understanding the genetic background of follicular tumors, and the previous literature is well cited and comprehensive.

Response: Thank you for this very kind statement concerning our paper and an appreciation of its value.

Comment: Only the following point need to be added to Results and Discussion. Regarding LOH in follicular carcinoma, differences by tumor size, presence of capsular invasion, multifocality, and nodal involvement should be shown.

Response: We would like to thank the reviewer for this suggestion of improving paper’s clinical value. We have added an information about the loss of heterozygosity (LOH) differences in patients with follicular thyroid cancer by tumor size, presence of capsular invasion, multifocality, and nodal involvement.

Reviewer 2 Report

In the paper “Follicular Thyroid Adenoma and Follicular Thyroid Carcinoma 2 – A Common or Distinct Background? LOSS of Heterozygosity 3 in Comprehensive Microarray Study”, the authors describe, through an array analysis in patients, the differences and similarities in LOH between FTA and FTC.

-       Page 5, in the results section, I would replace Table 2 in the supplementary Figures format of a better-structured Excel table to be included in the supplementary materials. Then, in place of table 2, in the results section, it would be better to convert the result of the comparison analysis into a Venn diagram.

-       The paper is very interesting, but the conclusion regarding new biomarkers that differentiate FTA from FTC still needs to be obtained. Arrays return too much information, so to be useful clinically, the authors should identify which markers are significantly different between FTA and FTC, compare them with known ones in the literature, and then create a reduced panel (or table) of all interesting markers that should be validated later, perhaps in follow-up work.

Author Response

Reviewer #2:

Comment: In the paper “Follicular Thyroid Adenoma and Follicular Thyroid Carcinoma 2 – A Common or Distinct Background? LOSS of Heterozygosity 3 in Comprehensive Microarray Study”, the authors describe, through an array analysis in patients, the differences and similarities in LOH between FTA and FTC.

-       Page 5, in the results section, I would replace Table 2 in the supplementary Figures format of a better-structured Excel table to be included in the supplementary materials.

Response: Thank you for this very kind statement concerning our paper and an appreciation of its value. We have shortened and simplified the Table 2 from the results section.

Comment: Then, in place of table 2, in the results section, it would be better to convert the result of the comparison analysis into a Venn diagram.

Response: We have added a Venn diagram showing the result of the comparison analysis. We believe that it has highly improved the clarity of data presentation and would like to thank you the reviewer for this excellent concept.

Comment: The paper is very interesting, but the conclusion regarding new biomarkers that differentiate FTA from FTC still needs to be obtained. Arrays return too much information, so to be useful clinically, the authors should identify which markers are significantly different between FTA and FTC, compare them with known ones in the literature, and then create a reduced panel (or table) of all interesting markers that should be validated later, perhaps in follow-up work.

Response: We have identified which markers are significantly different between FTA and FTC, and which are similar, compared them with known ones in the literature, and created a reduced panel (another Venn diagram) of all interesting markers that should be validated later. We hope that, as suggested by the reviewer, it can be of high value for further research.

Reviewer 3 Report

In the present manuscript, Borowczyk et al. extended their previous NGS results performed on human follicular thyroid adenomas (FTA) and follicular thyroid carcinomas (FTC) (Borowczyk et al., 2019 IJMS), by comparing the genomic landscape of FTA and FTC in order to identify regions of loss of heterozygosity (LOH). They found that FTA and FTC shared the same genetic background with some differences such as LOH in 11p11.2p11.12 only in FTA patients and LOH in 12q24.11q24.13, overlapping FOXN4, MYL2, PTPN11 genes. mainly in FTC.

Even if the manuscript is well organized and potentially interesting, it is however mainly descriptive and speculative with no functional experiment. The overall analysis of the large amount of OncoScan data is absolutely well performed and convincing, but i) the number of samples (patients) is too small considering that FTA is the most common benign thyroid tumor and that FTC is the second more common thyroid cancer after papillary thyroid carcinoma; ii) no experiments are provided by the authors about the expression of some of 12q24.11q24.13 mapping genes (FOXN4, MYL2, PTPN11) to understand if the allelic copy still present in FTC samples is inactivated or functional. The first issue should be addressed by increasing the number of FFPE specimens from additional patients. The second issue should be addressed by in situ hybridization or immunohistochemistry on the same FFPE specimens. By this way the authors could understand if the LOH of such genes is simply a FTC marker or if they represent potential tumor suppressors in FTC.

If the authors will address the issues above highlighted, the manuscript could be suitable for publication on Cancers.

Author Response

Reviewer #3:

Comment: In the present manuscript, Borowczyk et al. extended their previous NGS results performed on human follicular thyroid adenomas (FTA) and follicular thyroid carcinomas (FTC) (Borowczyk et al., 2019 IJMS), by comparing the genomic landscape of FTA and FTC in order to identify regions of loss of heterozygosity (LOH). They found that FTA and FTC shared the same genetic background with some differences such as LOH in 11p11.2p11.12 only in FTA patients and LOH in 12q24.11q24.13, overlapping FOXN4, MYL2, PTPN11 genes. mainly in FTC.

Even if the manuscript is well organized and potentially interesting, it is however mainly descriptive and speculative with no functional experiment. The overall analysis of the large amount of OncoScan data is absolutely well performed and convincing, but i) the number of samples (patients) is too small considering that FTA is the most common benign thyroid tumor and that FTC is the second more common thyroid cancer after papillary thyroid carcinoma; ii) no experiments are provided by the authors about the expression of some of 12q24.11q24.13 mapping genes (FOXN4, MYL2, PTPN11) to understand if the allelic copy still present in

FTC samples is inactivated or functional. The first issue should be addressed by increasing the number of FFPE specimens from additional patients. The second issue should be addressed by in situ hybridization or immunohistochemistry on the same FFPE specimens. By this way the

authors could understand if the LOH of such genes is simply a FTC marker or if they represent potential tumor suppressors in FTC.

If the authors will address the issues above highlighted, the manuscript could be suitable for publication on Cancers.

Response: We are aware that the number of samples (patients) is small, and we did not provide any experiments about the expression of mapping genes to understand if the allelic copy still present in FTC samples is inactivated or functional.

We consider this paper as a starting point for future research. The method which we used enables comprehensive LOH study which have not been performed previously. As far, studies on such patients and applied methodology are very limited - researchers have concentrated on only few candidate chromosomal regions in their papers. Analyzed LOH locations were targeted to known regions, whereas in our study we performed a comprehensive genome-wide LOH analysis, detecting novel relevant genetic loci and attempt to explore the complex interplay of distinct genetic alterations. To the best of our knowledge, researchers so far did not use the OncoScan™ microarray assay (Affymetrix, USA). We found also that previous LOH study in FTA and FTC included a median number of 23 patients, therefore we found scientifically desirable and valuable to expand molecular study on additional patient cohort.

When we calculate minimum number of necessary samples to meet the desired statistical constraints, assuming alpha at 0.05 we see that the number of samples may be consider sufficient.

However, we would like to thank the reviewer for raising this point and we decided to add the description of study limitation to the Discussion section which is as follow:

The study has two main limitations. The first limitation is the small number of participants. However, the method used in our study - highly multiplexed molecular inversion probes for SNP loci, to the best of our knowledge, has been used for the first time in FTA and FTC analysis. As opposite to previous research (also including small number of patients), we were able to simultaneously detect many genetic changes. The second one is a lack of any experiments about the expression of mapping genes to understand if the allelic copy still present in FTC samples is inactivated or functional. It requires a further analysis of by in situ hybridization or immunohistochemistry on the same FFPE specimens to understand if the LOH is simply FTC marker or if they represent potential tumor suppressors in FTC. We consider our study as a starting point for future research increasing the number of FFPE specimens from additional patients and including functional analysis.

Round 2

Reviewer 3 Report

"Cancers" is a highly impacted journal and one of the more read journal in the field of cancer. In my opinion, the analysis of the authors in the present manuscript, even if well performed and well described, is restricted and limited to a small number of patients and it is not suitable as a starting point. For this reason, an IHC or ISH could improve and strenghten the results presented.